# [Re] Solving Phase Retrieval With a Learned Reference

## Reproducibility Summary

### Scope of Reproducibility

This report reproduces the experiments and validates the results of the ECCV 2020 paper "Solving Phase Retrieval with a Learned Reference" by Hyder et al. [8]. The authors consider the task of recovering an unknown signal from its Fourier magnitudes, where the measurements are obtained after a reference image is added onto the signal. In order to solve this task a novel, iterative phase retrieval algorithm, presented as an unrolled network, that can train a such reference on a small amount of data is proposed. It is shown that the learned reference generalizes well to unseen data distributions and is robust to spatial data augmentation like shifting and rotation.

### Methodology

We use the provided original code to reproduce the experiments from Hyder et al. [8] that validate the proposed claims. Nevertheless, we refactor the code base to accelerate the performance and we extent it to carry out experiments where no code is available. We perform a hyperparameter search to investigate the influence and optimal values of the learning rates in both the training and retrieval process. Additionally, we do an ablation study to evaluate the necessary parts of the proposed algorithm. For our experiments we use a single NVIDIA TESLA P100 GPU with 16GB RAM and approximately 100 computational hours for all experiments together.

### Results

In general, we are able to reproduce the results of Hyder et al. [8]. Because of the hyperparameter search, we are certain that the results are not cherry-picked and mostly reproducible using the authors' implementation of the algorithm. With our additional experiments, we further strengthen the validity of the proposed method and help future researchers and practitioners by providing additional information on the learning rates in the training and retrieval process.

### What Was Easy

The authors provide an implementation of their algorithm that is executable in our environment after exchanging deprecated functions. The considered datasets are open access, hence easy to use. Furthermore, the computational cost is fairly low such that we could run extensive experiments and even compare different hyperparameter settings.

### What Was Difficult

We spend some effort to understand the authors' implementation, as it is marginally documented and the used computational tricks are not explained in detail. Moreover, it contains some redundant code which slows down computation. Beyond refactoring, we had to extent the implementation to be able to run our experiments. The lack of information about the learning rates slowed down the reproduction of the results, as we first had to investigate the influences on the training and retrieval process before we could adjust the parameters effectively.

### Communication With Original Authors

We were in contact with the authors via mail and we would like to thank the authors for helping us. Especially, we thank Rakib Hyder who kindly answered all our questions regarding implementation details and hyperparameters and Salman Asif who was open for our implementation suggestions and provided useful feedback for this report.

# 1  Introduction

Many optical detection devices can only measure the Fourier magnitude of a signal (e.g., the intensity of light) but not its Fourier phase. This systematic loss of information is known as the phase problem and often arises in X-ray crystallography [12], microscopy [17], astronomical imaging [5] and coherent diffraction imaging [2]. The goal of phase retrieval algorithms is to efficiently recover the phase of a signal from its phaseless magnitude measurements. A special problem instance is Fourier phase retrieval, where amplitudes of a Fourier transformed signal are measured and the task is to recover the original real or complex valued signal.

In general, there is no unique mapping from the magnitude to the target signal, thus there exist various approaches to solve it. Mainly inspired by solving holographic phase retrieval using a reference signal by Barmherzig et al. [1], the authors apply a similar approach to Fourier phase retrieval. Therefore, they assume a setting where the target signal $x$ and the reference signal $u$ are additive and overlapping, i.e.,

$$y = |F(x) + F(u)| + \eta, \tag{1}$$

where $F$ is the n-dimensional Fourier transformation and $\eta$ is the measurement noise. For this particular setting, Hyder et al. [8] propose a novel, data-driven retrieval algorithm as an unrolled network with a fixed number of layers. It is capable to learn a reference signal $u$ and subsequently solve the phase retrieval problem utilizing $u$ to recover the target signal $x$ solely from the measurements $y$.

# 2  Scope of Reproducibility

In this paper we reproduce the most important experiments using the method proposed by Hyder et al. [8]. We examine, refactor and extend the original code which we incorporate into our scripts to run our experiments.

## 2.1  Addressed Claims From the Original Paper

We validate in this paper the following claims from Hyder et al. [8]:

- The presented iterative algorithm is able to learn a reference signal and can utilize it in Fourier phase retrieval to improve the recovery of the target signal. Moreover, it requires only a small amount of training data to learn a reference.
- The learned reference is (i) robust to data augmentation in spatial space, (ii) it generalizes well to unseen data distribution and (iii) it is better than other types of references, e.g., random references.

## 2.2  Our Contribution

Our contributions in this report are:

1. We redo the experiments on phase retrieval with a learned reference with all datasets and report all used parameters.
2. We reproduce the generalization study with a subset of the data and report all used parameters.
3. We validate the robustness claims with our experiments and use furthermore an additional dataset.
4. We reproduce the experiments on the benefits of a learned references and also extend them with further types of references and new images.
5. We validate and extend the comparison with some baseline phase retrieval algorithms.
6. We perform an extensive hyperparameter search to analyze the influence of the learning rates on the reconstruction. We show that the performance of the algorithm can be improved by tuning the learning rates.
7. We investigate on the necessity of a reference and on the amount of oversampling in the training and recovery process.

# 3  Methodology

Mainly, we use the Algorithm 1 and 2 from [8] which are implemented in PyTorch [14] to validate the proposed claims and we mostly follow the restrictions and approaches described in the paper.

## 3.1 Model Description

In order to reconstruct the target signal $x^*$ given a reference signal $u$ and measurements $y = |F(x^*) + F(u)|$, Hyder et al. [8] propose to minimize the loss function

$$L_x(x; y, u) = \|y - |F(x) + F(u)|\|_2^2 \tag{2}$$

using a gradient descent algorithm

$$x^{k+1} = x^k - \alpha \nabla_x L_x(x^k; y, u), \tag{3}$$

where $\alpha > 0$ is the learning rate and $x^k$ is the reconstruction of the $k$-th iteration (with $x^0$ being properly initialized). The authors interpret the $K$ iterations as an unrolled network with $K$ layers, such that each layer of the network represents a single gradient descent update step. So, the input to the network is $y$ and $u$ and the output can be written as a function $x^K(y, u)$.

The reference signal $u$ is learned from a training dataset of images $x_1, \ldots, x_N$ and corresponding measurements (magnitudes) $y_1, \ldots, y_N$ for a given reference $u$, which could be written as

$$y_i = |F(x_i) + F(u)|. \tag{4}$$

Since for the training images and their magnitudes are known, a good reference image $u$ can by learned by minimizing the least-squares error

$$L_u(u; x_1, \ldots, x_n, y_1, \ldots, y_n) = \sum_{i=1}^{N} \|x_i - x^K(y_i, u)\|_2^2 \tag{5}$$

between signals from the training dataset $x_1, \ldots, x_N$ and their corresponding reconstructions $x^K(y_1, u), \ldots, x^K(y_N, u)$ using the unrolled network, Eq. (3).

This loss is minimized by gradient descent

$$u^{j+1} = u^j - \beta \nabla_u L_u(u^j; x_1, \ldots, x_n, y_1, \ldots, y_n), \tag{6}$$

where $\beta > 0$ is the learning rate for the reference and $u^j$ is the reference in the $j$-th iteration (with $u^0$ being properly initialized). The gradient $\nabla_u L_u$ can be calculated via backpropagation. The update rule Eq. (6) is applied for fixed number of iterations $J$.

## 3.2 Datasets

Throughout our experiments, we use the same datasets as in the original work [8], i.e., MNIST [10], EMNIST [3], FMNIST [16], CIFAR-10 [9], SVHN [13], CelebA [11] and also 6 additional standard benchmark images [1]. Three of these images were also used in the original work [8] and three are new.

Mainly, we access the data via provided code by the authors. For training a reference, we use always 32 images from the training datasets and we test on the same amount of data as proposed by Hyder et al. [8]: We use 10000 test images from MNIST, FMNIST and CIFAR-10, 24800 for EMNIST, 26032 from SVHN and 1000 from CelebA, if not mentioned otherwise. Furthermore, our preprocessing pipeline is similar to the original work [8]: All used images are converted to greyscale, have intensity values in range $[0, 1]$ and we reshape images from MNIST, EMNIST, FMNIST, CIFAR-10, SVHN to $32 \times 32$, images from CelebA to $200 \times 200$ and the standard benchmark images to $512 \times 512$.

## 3.3 Hyperparameter

According to [8], we restrict the intensity values of the reference signal $u$ to be within the interval $[0, 1]$ throughout all experiments. Furthermore, we oversample four times in spatial domain by padding the input image with a black border, as this makes the problem more well-behaved. Additionally, our unrolled network always consists of 50 layers and we consider a noise free setting for training and retrieval. However, we provide detailed parameter configurations for all our experiments in the results section of the respective experiment.

## 3.4 Experimental Setup

To run the original code, we replaced deprecated functions from the algorithm and imported MNIST and CelebA manually. We use PyTorch 1.5.0 [14], scikit-image 0.18.1 [15] and NumPy 1.21.0 [7] as environment and conduct our experiments in Jupyter notebooks. To compare our results with the original ones, we mainly focus on the peak-signal-noise-ration (PSNR) over the test images. The used code is available on GitHub [2].

---

[1] `https://homepages.cae.wisc.edu/~ece533/images/` (Accessed on June 25, 2021)

[2] `https://anonymous.4open.science/r/Machine_Learning_Reproducibility_Challenge_Spring_2021-3910/`

| Dataset | Hyder et. al. [8] | Our reproduced results | |
| | | Provided reference | Our trained reference |
| --- | --- | --- | --- |
| MNIST | 66.54 | $66.54 \pm 24.15 \; (\alpha = 1.348)$ | $66.53 \pm 14.98 \; (\alpha = 1.177)$ |
| EMNIST | 58.72 | $58.73 \pm 15.71 \; (\alpha = 1.010)$ | $58.71 \pm 19.31 \; (\alpha = 1.160)$ |
| FMNIST | 57.81 | $57.83 \pm 13.64 \; (\alpha = 1.052)$ | $57.88 \pm 19.36 \; (\alpha = 1.320)$ |
| SVHN | 57.51 | $57.50 \pm \; 9.66 \; (\alpha = 1.520)$ | $57.55 \pm 11.58 \; (\alpha = 1.660)$ |
| CIFAR-10 | 41.60 | $41.61 \pm 12.37 \; (\alpha = 1.315)$ | $41.68 \pm 12.78 \; (\alpha = 1.720)$ |
| CelebA | 39.00 | $39.12 \pm 10.78 \; (\alpha = 1.400)$ | $39.06 \pm 11.21 \; (\alpha = 1.870)$ |

Table 1: Comparison of mean PSNR values reported in the original work [8] and reproduced results using the provided reference and references that were trained from scratch. The learning rates were tuned so that our results match the reported values from the paper.

## 3.5 Computational Requirements

The original implementation requires a GPU with CUDA. Therefore, we use a single NVIDIA TESLA P100 GPU with 16 GB memory for our experiments. Overall, we used approximately 100 GPU hours but it is possible to verify the proposed claims within about 3 GPU hours, if all parameters are known. Moreover, by finding and removing unused code we are able to decrease the runtime of the algorithm by 15 to 30 times, depending on the shape of the image. For example, retrieving 26032 images with shape $32 \times 32$ takes approximately 9 seconds instead of 180 seconds.

# 4 Results

## 4.1 Reconstruction Using Learned References

In our first experiment we reproduce the mean PSNR values on MNIST, EMNIST, FMNIST, SVHN, CIFAR-10 and CelebA that are reported in Fig. 2 of [8], see our Tab. 1. We use the provided pre-trained references and additionally self-trained references and compare the mean peak-signal-noise-ratio (PSNR) values as the performance criterion. For matching results we tune both $\beta$ (the learning rate for the reference $u$) [3] and $\alpha$ (the learning rate for the recovery) in the training and reconstruction process. We explain these hyperparameters more detailed in Sec. 4.6. However, in reconstruction we keep $\beta = 1$ fixed and provide the $\alpha$ values used in the retrieval process additional to the results also shown in Tab. 1.

By adjusting the learning rate $\alpha$ in the recovery process, we are able to reproduce all reported mean PSNR values within a deviation of $1\%$ using the provided references and also our self-trained references. For MNIST, EMNIST and FMNIST we train for 5 epochs with $\alpha = 1$ and $\beta = 1$, for CelebA we need to train for at least 15 epochs with the same learning rates. To reproduce the reported mean PSNR for CIFAR-10 we set $\alpha = 1.3$ during training and train for 5 epochs. For SVHN we need to set $\alpha = 1.3$ and $\beta = 10$ while we train for 10 epochs to receive the reported mean PSNR values.

## 4.2 Generalization Study

We verify that our self-trained references also have a generalization property by reproducing a subset of the original generalization study from [8]. We use MNIST, FMNIST and CIFAR-10 as a representation for each type of images, i.e., artificial and real-world images. Our reproduced results are presented in Tab. 2. We find that with our self-trained references all reported values except for one are reproducible within $1\%$ deviation by tuning $\alpha$ in reconstruction. Nevertheless, recovery of CIFAR-10 test images with a self-trained FMNIST reference results in a maximum mean PSNR of 33.75dB using $\alpha = 1.855$ but Hyder et al. [8] report 42.85dB instead. With the provided FMNIST reference, we obtain only a maximum mean PSNR of 40.72dB using $\alpha = 1.870$ (found via hyperparameter search).

Additionally, we examine the same experiment with fixed learning rate $\alpha = 1$ in the recovery process to investigate if the described trends of the references behavior, hold for our self-trained references as well. We present our experimental results in Tab. 3.

While MNIST and FMNIST references are reasonable reference signals for each other, the performance drops on CIFAR-10 which supports the observation of the authors. In contrast, the CIFAR-10 reference is more valuable for the other datasets than for itself while this is not the case in the original study. Moreover, it performs better than reported by

---

[3]Note: $\beta$ is called `lr_u` in the implementation provided by the authors.

| Trained on | Evaluated on | | |
| | MNIST | FMNIST | CIFAR-10 |
|---|---|---|---|
| MNIST | $66.53 \pm 14.98$ ($\alpha = 1.177$) | $40.62 \pm 12.66$ ($\alpha = 0.795$) | $31.71 \pm 9.00$ ($\alpha = 0.950$) |
| FMNIST | $40.75 \pm 14.45$ ($\alpha = 0.730$) | $57.88 \pm 19.36$ ($\alpha = 1.320$) | $40.72 \pm 16.93$ ($\alpha = 1.870$) |
| CIFAR-10 | $31.76 \pm 8.31$ ($\alpha = 0.405$) | $36.45 \pm 9.30$ ($\alpha = 0.550$) | $41.68 \pm 12.78$ ($\alpha = 1.720$) |

Table 2: Comparison of mean PSNR of the generalization study using tuned learning rate $\alpha$. Again, the learning rates were tuned so that our results match the reported values from the paper.

| Trained on | Evaluated on | | |
| | MNIST | FMNIST | CIFAR-10 |
|---|---|---|---|
| MNIST | $59.76 \pm 13.27$ | $45.77 \pm 15.31$ | $32.07 \pm 9.26$ |
| FMNIST | $49.44 \pm 18.11$ | $49.07 \pm 15.16$ | $28.58 \pm 11.65$ |
| CIFAR-10 | $52.04 \pm 14.26$ | $49.63 \pm 15.20$ | $37.20 \pm 9.89$ |

Table 3: Comparison of mean PSNR of the generalization study using fixed learning rate $\alpha = 1$ in recovery.

Hyder et al. [8] as it is even better than the FMNIST reference on FMNIST. In conclusion, we observe slightly different behaviour in our experiments but overall, the learned references generalizes well, as claimed in the paper.

### 4.3 Robustness to Data Augmentation

These experiments validate that our self-trained references are robust against shifts, flips and rotations in the spatial domain as it is reported in [8]. We use MNIST and CIFAR-10 for reproduction according to the authors' choice and SVHN as an additional dataset. Throughout the experiment, the learning rate in reconstruction is fixed to $\alpha = 1$ and we evaluate our experiment only on 1000 test images from each dataset. A summary of our results is presented in Tab. 4.

While we observe that flipping and rotating in the spatial domain barely decrease the mean PSNR on all evaluated datasets, only MNIST is fairly robust to shifting. Hence, for SVHN the mean PSNR drops by 29% while for CIFAR-10 it falls off by nearly 40%. That means, their recovery results are equal or worse than the results using a random reference. We consider the loss of information from shifting with the associated zero padding to be the cause for this, as it has less impact on the dark-edged MNIST images. However, since Hyder et al. [8] also show a decreased mean PSNR for shifting in Fig. 4 of their paper, we can validate their results.

### 4.4 On the Benefit of a Learned Reference

With this experiments we evaluate the advantages of a learned reference against (i) a constant, (ii) a randomly sampled and (iii) a handcrafted reference. We consider the six standard benchmark images. As references we use our self-trained CelebA and CIFAR-10 references, which we resize to $512 \times 512$ by upscaling. The parameters are fixed in reconstruction to $\alpha = 1.92$ (for best mean PSNR in recovery). Fig. 1 shows our experimental reconstructions of the benchmark images together with the achieved PSNR values.

First, we can show that the reported results from Hyder et al. [8] are reproducible, as we receive similar reconstruction results with our self-trained CelebA reference. Additionally, we repeat the experiment with our self-trained CIFAR-10 reference but only obtain reconstruction results between the result using a random and the CelebA reference.

To generate our random references we follow the description in [8], i.e., we draw from a uniform distribution with range $[0, 1]$. Additionally, our random reference is drawn with shape $30 \times 30$ and resized to $512 \times 512$, because this setup performs best. Finally, we report the results of the best performing reference from 100 randomly sampled references also in Fig. 1. We observe that our experimental results are similar to the original reconstructions results.

| Dataset | No augmentation | Shift (5 pixel left and up) | Flip | Rotation (90° clockwise) |
|---|---|---|---|---|
| MNIST | $59.59 \pm 13.33$ | $60.89 \pm 14.11$ | $49.71 \pm 17.09$ | $49.65 \pm 17.43$ |
| CIFAR-10 | $47.27 \pm 10.14$ | $28.48 \pm 11.73$ | $47.10 \pm 9.88$ | $41.18 \pm 13.00$ |
| SVHN | $37.04 \pm 9.94$ | $26.50 \pm 7.20$ | $38.08 \pm 9.24$ | $38.13 \pm 10.13$ |

Table 4: Analysis of the robustness to different data augmentation methods. Results are reported in mean PSNR with standard deviation.

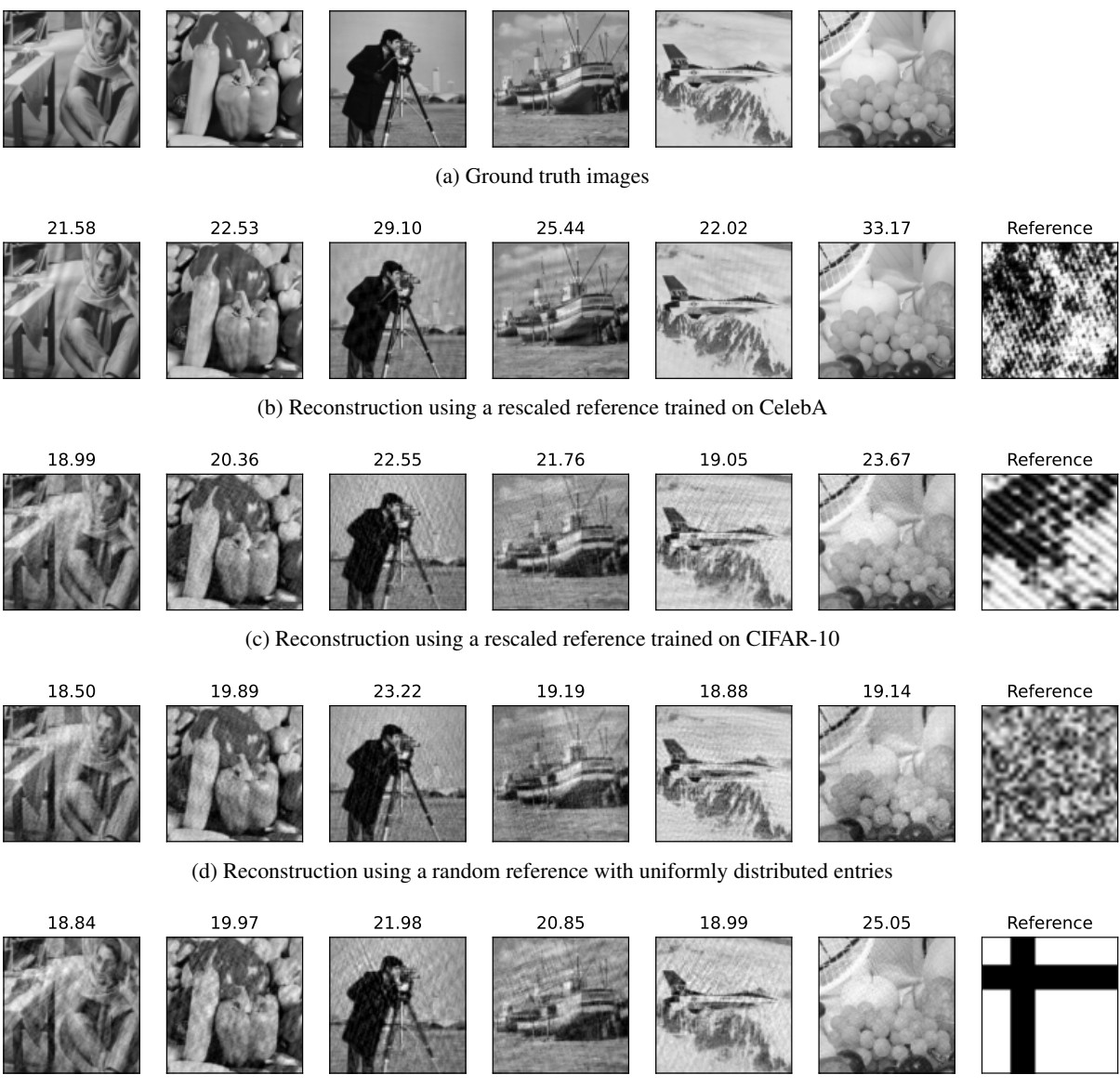

(a) Ground truth images

21.58  22.53  29.10  25.44  22.02  33.17  Reference

(b) Reconstruction using a rescaled reference trained on CelebA

18.99  20.36  22.55  21.76  19.05  23.67  Reference

(c) Reconstruction using a rescaled reference trained on CIFAR-10

18.50  19.89  23.22  19.19  18.88  19.14  Reference

(d) Reconstruction using a random reference with uniformly distributed entries

18.84  19.97  21.98  20.85  18.99  25.05  Reference

(e) Reconstruction using a handcrafted reference

Figure 1: Reconstruction results on benchmark images using different references (PSNR on top). From top to bottom: ground truth, our trained CelebA reference, our trained CIFAR-10 reference, best random reference (uniform distributed, evaluation on 100 references per image), best handcrafted reference.

To show the advantage against a flat reference, we consider different flat references (all entries set to the same value), where we obtain comparable results for different flat references. Similar to the observation of the authors, the recovery results are frequently worse than results obtained with a random reference. We observe minor improvement of some decibel in mean PSNR if we assemble squares or lines manually to common figures like crosses, without any relation to the content of the pictures. However, the reconstructed images are still less noisy if we use a random reference as shown in Fig. 1. Overall, we can validate the reported results from [8], in particular the learned reference performs best against all other evaluated types.

## 4.5 Comparison With Baseline Algorithm

In this section, we validate the reported results of the hybrid-input-output algorithm (HIO) [4] and extend the experimental evaluation by including two more baseline phase retrieval algorithms: Fienup's input-output and the Gerchberg-Saxton (GS) algorithm [6]. We re-implement all three algorithms from scratch using NumPy [7]. We oversample the test images four times in spatial domain and run the algorithms for 100 iterations on each image with a step size of $\beta = 0.8$ for input-output [4] and HIO [4]. Also, the reconstructions are clipped to intensity values in range

| | Algorithm | MNIST | EMNIST | FMNIST | SVHN | CIFAR-10 |
|---|---|---|---|---|---|---|
| Ours | Input-Output | $9.80 \pm 1.35$ | $9.85 \pm 1.46$ | $8.74 \pm 2.63$ | $6.68 \pm 1.85$ | $7.80 \pm 1.73$ |
| | GS | $9.82 \pm 2.44$ | $9.99 \pm 2.41$ | $11.25 \pm 3.63$ | $17.89 \pm 3.77$ | $16.34 \pm 3.08$ |
| | HIO | $10.53 \pm 3.81$ | $10.81 \pm 3.93$ | $14.06 \pm 8.54$ | $31.90 \pm 16.45$ | $28.33 \pm 13.92$ |
| Hyder et al. [8] | HIO | 9.04 | 8.42 | 9.65 | 19.87 | 14.70 |

Table 5: Comparison of mean PSNR values (with standard deviation) by the baseline methods without use of a reference signal. Additionally, to the results of the HIO algorithm, we report the results for the input-output and the GS algorithm.

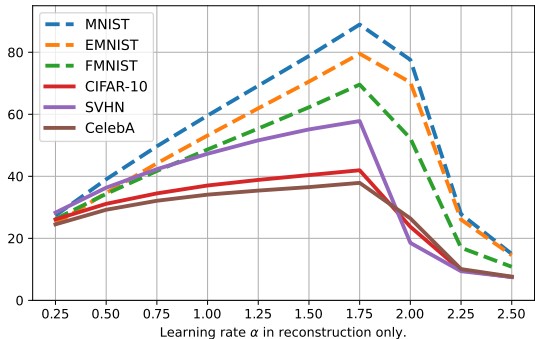

Figure 2: Results from the hyperparameter search for variable learning rate $\alpha$ in reconstruction.

$[0, 1]$. For each image, we select the best PSNR from the cropped reconstruction and the cropped, flipped and shifted one. Tab. 5 shows the results on the different datasets. Overall, we can validate the claim by Hyder et al. [8], even though our HIO [4] implementation performs slightly better than the one reported in the original work.

## 4.6 Hyperparameter Search

Since we have no access to the original learning rates, we perform an extensive grid search on the hyperparameters $\alpha$ and $\beta$. In this study we use 5 epochs during training and evaluate on 1000 images.

We start with the learning rate $\alpha$ which is used to update the reconstruction in training a reference and also in the retrieval process. For this, we use the self-trained references and keep $\beta = 1$ fixed while $\alpha$ is variable in recovery. Our results on all used datasets are presented in Fig. 2. Surprisingly, there is a general increase of the mean PSNR among all datasets for rising $\alpha$ values up to a peak in range $\alpha \in [1.75, 2.00]$. Unfortunately, also the standard deviation grows proportional to the higher mean PSNR values. Nevertheless, these effects are stronger on artificial images than on real-world images.

For our second experiment, we train with variable $\alpha$ on a logarithmic scale while we keep $\beta = 1$ fixed in training and fix $\alpha = 1$ in the recovery process. Fig. 3a shows our results. Among the considered datasets SVHN has the smallest range but provides still valuable reconstructions for $\alpha \in [0.1, 1]$. However, for all datasets, an extensively small or big $\alpha$ leads to learning a worse reference than a randomly sampled one, while the best recovery results are mainly in $\alpha \in [0.1, 1]$.

Finally, we train with a variable reference learning rate $\beta$, while we keep $\alpha = 1$ fixed. Our results on a representative subset are shown in Fig. 3b. In general, choosing small value for $\beta$ leads to learning useless references. Nevertheless, we observe no general pattern for optimizing the retrieval performance by adjusting $\beta$ in training but valuable results often ranges in the interval $\beta \in [0.1, 10]$.

## 4.7 Ablation Study

For our ablation study we investigate whether a reference is really necessary for the retrieval process and study how oversampling in spatial domain influences the reconstruction quality. For this experiment, we use MNIST, CIFAR-10 with 1000 test images as well as the common "cameraman" image in shape $512 \times 512$. The learning rates are fixed to $\alpha = 1$ and $\beta = 1$.

First, we run the reconstruction algorithm without using a reference. We observe that the mean PSNR decreases drastically, e.g., for MNIST the mean PSNR is $8.92$dB. We observed similar results for other datasets such that we can conclude that a reference is required to obtain reasonable reconstructions.

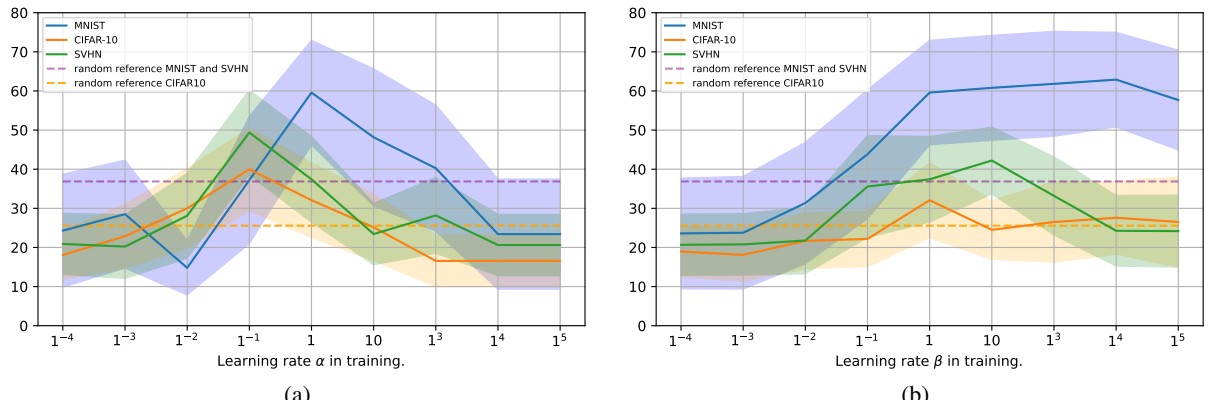

Figure 3: Results from the hyperparameter search for the learning rates: (a) mean PSNR of reconstructed images with with references trained using different learning rates $\alpha$ and (b) mean PSNR of reconstructed images with references trained using different learning rates $\beta$. During reconstruction a fixed learning rate $\alpha = 1$ has been used.

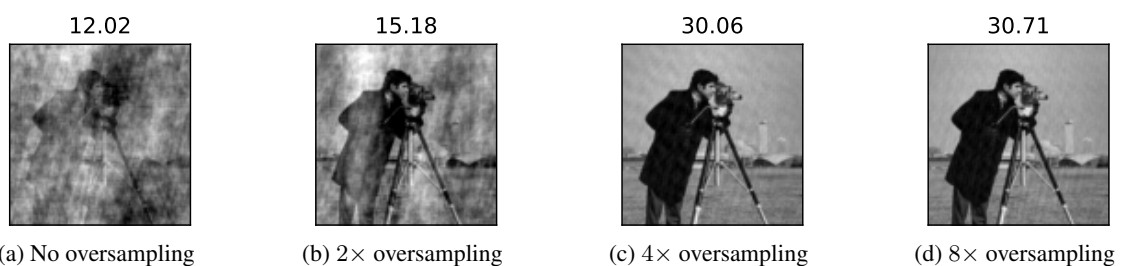

(a) No oversampling    (b) $2\times$ oversampling    (c) $4\times$ oversampling    (d) $8\times$ oversampling

Figure 4: Reconstructions results with CelebA reference (trained with oversampling) and use of different amount of oversampling during reconstruction (mean PSNR values on top of the images).

Second, we use a reference that was trained with $4\times$ oversampling and we vary the amount of oversampling during the recovery process. Fig. 4 shows our results for a single benchmark image. We observe, that using no oversampling or $2\times$ oversampling during reconstruction leads to cloud-like artifacts. Oversampling $4\times$ in recovery is successful. Oversampling by a factor of $8$ leads only to marginally improved performance.

Additionally, we find that we can obtain reasonable reconstructions with references that were trained without any oversampling, if we use $4\times$ oversampling in the retrieval process. For example, using this approach we receive a mean PSNR of $47.90$dB on MNIST which is just $6.38$dB PSNR below the result with a reference that was trained using $4\times$ oversampling. Therefore, it might be a consideration to omit oversampling while training a reference, as it is a trade-off between reconstruction quality and computational requirements.

## 5 Discussion

In conclusion, we can verify that the unrolled network proposed by Hyder et al. [8] is capable of learning a valuable reference that can be utilized to recover a signal from its Fourier magnitude measurement. We trained our references from scratch and we demonstrated that they are similar enough to the original ones. Moreover, we encountered no major contradiction in our experiments if we use new data, references or generative methods. However, an extensive hyperparameter search was necessary to match the reported results. Also, the hyperparameter search reveals that one should focus on tuning the learning rate $\alpha$ during reconstruction as it yields to performance improvements across all datasets. Our ablation study shows that oversampling during training can be omitted to save computational resources.

Nonetheless, by providing an official implementation of their algorithm the authors enabled future researchers to utilize their method. Furthermore, we are grateful to the authors for kindly answering all of our questions regarding the implementation and providing feedback on our results.

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
