# OpenReview forum: "[Re] Solving Phase Retrieval With a Learned Reference"
_ML_Reproducibility_Challenge/2021/Fall — RC2021_

### Official Review · Reviewer_fWcK · 2022-02-25
**Confirmation of results and that a reference image is important**

**Rating:** 7
**Confidence:** 4

**Review:**

The report summarizes the problem statement of the original paper "Solving Phase Retrieval with a Learned Reference" as recovering a signal from its Fourier amplitudes. Only later in the paper one can guess that the Fourier transform is applied to images, but that should be mentioned in the summary at the beginning.
Source code for the experiments in the original paper was available and used for the reproducibility study, but it was also refactored for performance improvements and extended by the authors. It is available online and nicely commented in the form of Jupyter notebooks. By communicating with the original authors the authors obtained additional information about implementation details and hyperparameters.

The main experiments of the original paper were reproduced successfully. As learning rates were unknown a hyperparameter search was performed on a grid in order to find their optimal values. In an ablation study the authors tested if a reference image is really needed, because that is the main defining feature of the original algorithm. It turns out that a reference is necessary, but oversampling during training can be omitted in order to speed it up.

The report is well written and clearly discusses the results of the reproducibility study. I recommend to make an explicit recommendation to the original authors for improving reproducibility.

---

### Official Review · Reviewer_J7NU · 2022-03-07
**A good reproducing report**

**Rating:** 8
**Confidence:** 3

**Review:**

This paper reproduced the original paper based on the code provided by the original authors and validated the original claims.
This reproducing paper studies the influence of hyperparameters, especially the learning rate parameters for the training and inference process. The authors also conducted new ablation studies to evaluate the importance of the parts of the method. The authors had proper communications with the original authors during the reproduction. Specifically, the authors improved the original code and made the computation more efficient (15 to 30 times fasters). The reproducibility summary and the whole report are well represented.

---

### Meta-Review · Area_Chair_jxie · 2022-04-08

**Recommendation:** Accept
**Confidence:** 5

**Metareview:**

The reviewers agree that the paper is a well written, good effort in reproduction. The paper also provides additional ablation studies which is helpful for the understanding of the main paper. Overall the quality is high, which warrants acceptance in this venue.

---

### Decision · Program_Chairs · 2022-04-09

**Decision:**

Accept

**Comment:**

Following the recommendation of reviewers and meta-reviewer, the paper is accepted for ML Reproducibility Challenge 2021, and will be published in the upcoming special edition of ReScience Journal.